# Dysregulated Liver Metabolism and Polycystic Ovarian Syndrome

**DOI:** 10.3390/ijms24087454

**Published:** 2023-04-18

**Authors:** Muhammad Sohaib Khan, Hee-Sun Kim, Ranhee Kim, Sang Ho Yoon, Sang Geon Kim

**Affiliations:** 1College of Pharmacy and Integrated Research Institute for Drug Development, Dongguk University-Seoul, Goyang-si 10326, Republic of Korea; muhammadsohaibkhan786@gmail.com; 2Department of Obstetrics and Gynecology, Dongguk University Ilsan Medical Center, Goyang-si 10326, Republic of Korea; smallkong@dumc.or.kr (H.-S.K.); ranhee.kim@dumc.or.kr (R.K.); yoonprou@gmail.com (S.H.Y.); 3Department of Obstetrics and Gynecology, Dongguk University Medical College, Goyang-si 10326, Republic of Korea

**Keywords:** liver-to-ovary axis, metabolic mechanisms, hyperglycemia, liver-secretory protein, infertility

## Abstract

A significant fraction of couples around the world suffer from polycystic ovarian syndrome (PCOS), a disease defined by the characteristics of enhanced androgen synthesis in ovarian theca cells, hyperandrogenemia, and ovarian dysfunction in women. Most of the clinically observable symptoms and altered blood biomarker levels in the patients indicate metabolic dysregulation and adaptive changes as the key underlying mechanisms. Since the liver is the metabolic hub of the body and is involved in steroid-hormonal detoxification, pathological changes in the liver may contribute to female endocrine disruption, potentially through the liver-to-ovary axis. Of particular interest are hyperglycemic challenges and the consequent changes in liver-secretory protein(s) and insulin sensitivity affecting the maturation of ovarian follicles, potentially leading to female infertility. The purpose of this review is to provide insight into emerging metabolic mechanisms underlying PCOS as the primary culprit, which promote its incidence and aggravation. Additionally, this review aims to summarize medications and new potential therapeutic approaches for the disease.

## 1. Introduction

Polycystic ovarian syndrome (PCOS) is a common endocrinological disorder affecting 5–10% of reproductive women and is characterized by chronic anovulation and hyperandrogenism. According to the Rotterdam criteria formulated in 2003, PCOS is diagnosed when at least two ovulatory dysfunctions (OD), clinical or biochemical hyperandrogenism (HA), and polycystic ovarian morphology (PCOM) are detected by ultrasound scan [1,2]. OD means either irregular menses at intervals of less than 21 or more than 35 days, or less than eight menstruations in one year. Clinical HA can manifest as hirsutism, male pattern alopecia, and acne, while increased serum androgen levels can suggest biochemical HA. PCOM is diagnosed if there are 12 or more follicles in either ovary measuring 2–9 mm in diameter and/or an ovarian volume exceeding 10 mL (in an updated guideline published in 2018, the number of follicles was increased to 20). According to the Rotterdam criteria, four different phenotypes are classified as A (HA, OD, and PCOM), B (HA and OD), C (HA and PCOM), and D (OD and PCOM). PCOS can then be classified by phenotype as either “class PCOS” (phenotypes A and B), “ovulatory PCOS” (phenotype C), or “nonhyperandrogenic PCOS” (phenotype D) [2]. The diversity of the phenotypes means that the pathophysiology of PCOS is both complex and multifactorial.

Previously, PCOS was considered a reproductive disorder affecting fertility. However, PCOS has since been reclassified as a metabolic disorder since insulin resistance (IR) is acknowledged as one of the key pathogens [3,4]. Patients with PCOS commonly have a high prevalence of obesity and IR [3], carrying an increased risk of metabolic syndrome, type 2 diabetes (T2D), and cardiovascular disease [5,6]. More than half of PCOS patients are clinically obese. Studies conducted in the United States, Italy, China, and Spain showed that the incidence of obesity in PCOS patients was 69%, 38%, 43%, and 20%, respectively [7]. Centrally deposited body fat is usually observed in PCOS patients, with increased abdominal fat deposition particularly associated with HA and increased metabolic risk [1,5].

IR is defined as impaired insulin action in principal target organs such as muscle, fat, and liver. Although no consensus exists on how to evaluate the condition, a homeostasis model assessment (HOMA) for IR is considered to be a representative indicator [4,8]. Moreover, the cut-off values for HOMA for IR can depend on the researchers studying them. Researchers have already used the 66th, 90th, and 95th percentiles of the HOMA-IR index in a healthy population. In a Korean study using the HOMA-IR cut-off value, the prevalence of IR in PCOS women was 60.7% and 24.5% for the 75th and 95th percentiles, respectively [4]. The overall prevalence of IR among PCOS patients was between 30% and 35% and was greater in obese PCOS patients than in non-obese PCOS patients [1].

Obesity, central adiposity, and IR, in particular, are considered the main causes of PCOS. As one of the potential complications of obesity and associated metabolic disorders, infertility attracts particular attention because 15% of couples around the world suffer from fertility issues, which often correspond with obesity, dyslipidemia, and metabolic syndrome [9]. In particular, PCOS is associated with HA, OD, and PCOM, exhibiting the characteristics of irregular menstrual cycles and excessive androgen levels, including hirsutism and acne. Another study has reported that females with PCOS have a greater incidence of obesity, which can aggravate PCOS-related risk factors such as IR and metabolic disease [10]. In addition, an excess of androgens, which is one of the key features of PCOS, is considered to contribute to the incidence of PCOS [11].

The liver is the metabolic hub of the body because all ingested nutrients pass through it after intestinal absorption. Therefore, underlying molecular cascades of metabolic diseases include lipotoxicity, autophagy dysregulation, endoplasmic reticulum stress, IR, and other targets [12]. NAFLD is currently the most common liver disease globally and has been reported to affect 30% of people over the age of 18. Another study showed that NAFLD is strongly associated with obesity and T2D [12]. Recent studies show the high prevalence of NAFLD in PCOS patients [13]. Although hepatic steatosis is a benign condition and is also frequently found in female PCOS patients, advanced liver disease is often present in obese patients. The persistent hyperglycemic challenge posed by obesity or diabetes results in subsequent changes in liver metabolism that are not compensatory. These changes in liver metabolism are a common pathway shared by progressive diseases of the liver and ovaries. Therefore, PCOS patients must be screened for NAFLD by monitoring blood aminotransferase activity and using ultrasound to assess abdominal liver steatosis [14].

This review aims to provide insight into emerging liver metabolic mechanisms in PCOS as the primary culprit, which promote PCOS incidence and aggravation. Additionally, this study aims to summarize medications and potential therapeutic approaches for the disease.

## 2. Molecular Pathology

### 2.1. Epidemiology

According to the National Institute of Health (NIH), “classic” PCOS affects 6–10% of reproductive-age women and may be twice as high if using the broader Rotterdam criteria [15]. Data-monitored healthcare has estimated and forecasted the global prevalence of PCOS in females aged 15–49 years across the globe between 2019 and 2028 using historical prevalence data. Data-monitor healthcare obtained high-quality PCOS prevalence data from a systematic study and meta-analysis of the PCOS burden [16]. PCOS prevalence is determined in accordance with at least one set of diagnostic criteria from the NIH, the Rotterdam criteria, or the Androgen Excess and PCOS group (AE PCOS). Meta-analyses were carried out to show PCOS prevalence at global and regional levels, while regional estimates were reported for Asia, Europe, North America, and Oceania; when data for other areas were unavailable, global prevalence estimates were used [17].

Recent studies have shown the incidence of PCOS in morbidly obese individuals. The meta-analysis results, which included 2130 patients who underwent bariatric surgery, have revealed that the preoperative incidence of PCOS was 45.6%, with the incidence declining significantly to 6.8% after bariatric surgery and stabilizing at 7.1% after one year [18]. In another prospective observational study comprising 50 women who underwent bariatric surgery, 18 (36% of them) were diagnosed with PCOS [19]. The authors compared the clinical, hormonal, and radiological aspects of PCOS, revealing that after the surgery, their menstrual cycles normalized within 90 days, hirsutism resolved itself, mean serum testosterone levels decreased significantly, and metabolic syndrome resolved completely at the one-year follow-up. These studies show that bariatric surgery effectively treats PCOS in morbidly obese women (BMI over 40 kg/m^2^).

### 2.2. Clinical Signs and Pathology

Infertility is one of the most common chronic health issues affecting young adults. According to a clinical study, 57% of women have sought medical assistance for infertility issues, compared with 53% of males. On average, one in eight women and one in ten men aged between 16 and 74 experience fertility issues [20]. The diseases most often identified in women were PCOS, ovulatory disorders, unexplained infertility, endometriosis, pelvic adhesions, tubal occlusion, other tubal abnormalities, and hyperprolactinemia [21]. Another study categorizes reasons for infertility in females (45–55%) into ovulation problems (15–25%), fallopian tube and peritoneal issues (25–35%), and infertility of unknown etiology (10–20%) [22]. All women experience the menopausal transition, where estrogen synthesis stops. The absence of estrogen can pose health problems such as android adiposity (i.e., central or abdominal fat distribution), IR, dyslipidemia, and osteoporosis. As PCOS patients age, the ovarian cycle becomes normalized in their late reproductive years due to increasingly spontaneous ovulatory cycles. In a prospective cohort study comparing 118 “classic” female PCOS patients with age-matched healthy controls aged between 20 and 40, the proportion of anovulatory cycles decreased from 100% to 73% with the age of the patients [23]. The authors compared women with PCOS whose cycles became ovulatory with advancing age against those whose cycles remained anovulatory. The results showed that in women whose ovarian cycle normalized with advancing age, the metabolic indices of insulin sensitivity and lipid profile were similar to those seen in the controls. However, in women whose ovarian cycles became abnormal, the metabolic indices were worse than those of the controls. Even though the metabolic functions of some women with PCOS improve with age, as a consequence of menopause itself, changes in body shape and a preferential increase in visceral adiposity pose greater metabolic complications in PCOS patients, particularly obese patients. The key factor determining improvement in ovulatory cycles and further cardiovascular risk will need to be investigated in the future [24].

Women with PCOS are more likely to have dyslipidemia and IR with compensatory hyperinsulinemia, T2D, metabolic syndrome, and cardiovascular problems [25]. Moreover, PCOS exhibits similar characteristics to NAFLD, such as IR, T2D, dyslipidemia, decreased sex hormone-binding globulin (SHBG), increased aminotransferase activity, and xanthine oxidase levels [26]. In particular, xanthine oxidase (XO) is a pro-oxidant enzyme found in the liver and is a molybdopterin-flavoprotein with two similar subunits (145 kDa) [27]. XO and xanthine dehydrogenase are interchangeable forms of the same enzyme, xanthine oxidoreductase [28]. XO mainly regulates the deprivation of purines into uric acid, during which it produces two moles of superoxide and one mole of H_2_O_2_ [29]. The enhanced activity of XO leads to excessive xanthine oxidase-associated pro-inflammatory mediator activation, which further aggravates PCOS [30]. A meta-analysis revealed that the XO levels were significantly raised in PCOS patients [31]. Remarkably, Gα13 knockout mice also showed elevated levels of XO enzymes [32].

The pathophysiology and intrinsic processes of PCOS are complex since the etiologies vary and many symptoms are intricately linked [17]. The interaction of these systems causes clinical characteristics such as ovulatory dysfunction, hyperandrogenism, polycystic ovarian morphology, accompanying mood abnormalities, psychosexual dysfunction, and long-term morbidities. Several studies have demonstrated increased levels of depression and anxiety in women with PCOS [33]. The effects of PCOS are not only confined to quality of life issues, such as anovulation, infertility, and pregnancy complications, but also affect mental well-being, including depression and anxiety disorders [34].

### 2.3. Endocrinology

Ovulation is a biological reproductive procedure defined as the release of an oocyte from a ruptured dominant follicle in the ovary into the fallopian tube, where it may potentially be fertilized [35]. The ovary releases less than one percent of the total number of oocytes during the reproductive period. The number peaks at between six and seven million after 20 weeks in utero. The number of oocytes decreases to between one and two million at birth, with only approximately 300,000 oocytes remaining by puberty [36]. During the entire reproductive period of a woman, only 400 to 500 oocytes will actually be released. The normal ovarian cycle consists of three phases: the follicular phase, ovulation, and the luteal phase. The follicular phase usually lasts for ~10–14 days (although it is different for different women) and is the preparation period for ovulation. A single dominant follicle is selected from a group of immature follicles and begins to produce estrogen. Ovulation is achieved when the dominant follicle releases the oocyte. The luteal phase follows ovulation and lasts for 14 days (compared to the follicular phase, the length is relatively constant for every woman). The ruptured dominant follicle becomes a corpus luteum, producing progesterone and estrogen [37].

Follicular growth during the follicular phase requires gonadotropins from the anterior pituitary [38]. Follicle-stimulating hormone (FSH) and luteinizing hormone (LH) are known as gonadotropin hormones because of their impact on gonadal growth and function. Gonadotropin release from the anterior pituitary is regulated by gonadotropin-releasing hormone (GnRH) from the hypothalamus. As the follicles grow, they become known as preantral follicles (including primordial, primary, and secondary follicles) and then antral follicles (including tertiary and preovulatory follicles). Graafian follicles (also known as preovulatory follicles and then tertiary follicles) are the primary source of cyclic ovarian estrogen production [39]. Oocytes are present in the ovarian follicles and are surrounded by two types of somatic cells. Theca cells are found outside the basal lamina of the ovarian follicle and are involved in androgen production. Granulosa cells are found inside the basal lamina and are involved in the aromatization of androgen to produce estrogen.

Hypothalamic-pituitary-ovarian axis imbalance is believed to be a critical pathophysiology underlying PCOS, resulting in abnormally enhanced GnRH and LH pulse frequency in PCOS patients (Figure 1). Hypothalamic GnRH neurons are the main regulators of LH production. Physiologically, the GnRH pulse should be attenuated in response to follicular growth [40]. Since an oddly augmented GnRH pulse, which results in an excessive LH pulse in women with PCOS, has already been described, it is thought to be a causative event for PCOS development. An increased androgen effect on ovarian theca cells by excessive LH activity may lead to hyperandrogenemia, ovarian dysfunction, and metabolic disorders in PCOS patients [40].

Steroid hormone synthesis in the ovaries is based on the two-cell, two-gonadotropin theory, referring to the compartmentalization of steroid hormone production. Granulosa cells lack the enzymes required for steroidogenesis. Therefore, granulosa cells must receive androgen from theca cells to form the substrate for estrogen production [41]. As aromatases are abundant in granulosa cells and their activity is increased by FSH stimulation, the granulosa cells in growing follicles are the primary site for ovarian estrogen synthesis. By contrast, theca cells produce androgen in response to LH stimulation. LH secretion from the pituitary is regulated by a biphasic pattern by estrogen level [41]. At lower concentrations, estrogen inhibits LH secretion. However, at higher concentrations (>200 pg/mL), estrogen increases LH secretion, producing a midcycle LH surge. At ovulation, only the granulosa cells of the dominant follicle can express the LH/chorionic gonadotropin (CG) receptor (LHCGR) to respond to the LH surge, which is the signal to initiate ovulation.

Obese women that are not suffering from PCOS may exhibit HA features since adipose tissue, particularly in the abdominal area, has steroidogenesis enzymes that can produce androgen, and adipose tissue converts androstenedione to testosterone [42]. Additionally, because IR is present in obese patients, hyperinsulinemia contributes to HA through several mechanisms. (1) Insulin operates as a co-gonadotropin with LH on ovarian theca cells to boost ovarian androgen synthesis; (2) insulin enhances the adrenal response to ACTH for elevated adrenal androgen production; (3) insulin acts synergistically with LH/hCG-mediated ovarian follicle arrest to make PCOM; and (4) insulin induces hyperresponsiveness of the pituitary from GnRH to secrete an increased amount of LH in vitro studies and inhibits hepatic SHBG production, resulting in increased levels of free androgen [42]. Since insulin and androgen reduce hepatic SHBG synthesis, increased free androgen levels subsequently provoke further IR, ultimately resulting in a vicious cycle over time. It should be noted that HA in PCOS mainly results from excess androgen production in the ovaries. This accounts for 60% of cases, with the remaining 40% originating from the adrenals and adipose tissue. Anti-Müllerian hormone (AMH) is released by granulosa cells in follicles measuring less than 4 mm (preantral and antral follicles) [43]. It is frequently used to assess ovarian reserve, reflecting the number of antral follicles and primordial follicles. AMH levels are increased in PCOS patients since the PCOS ovary contains more follicles in the antral and preantral stages [44]. Higher levels of AMH in PCOS patients result in the loss of FSH-induced aromatase activity in granulosa cells, causing what is known as follicular arrest [45]. The results of several studies have shown that LH, FSH, insulin-like growth factor 1 (IGF1), and AMH all play a vital role in androgen conversion and thus contribute to PCOS resulting in oligo- or anovulation [46].

### 2.4. Sex Hormone Dysregulation

Jensen was the first researcher to put forward the theory that the liver stores steroidal hormones such as estrogen and testosterone for metabolism. This theory was later validated with the help of scientific experiments showing that the reason for the responsiveness of the liver to sex hormonal metabolism is due to the presence of androgen and estrogen receptors in the liver [47].

HA is one of the main clinical symptoms noted in women with PCOS and is caused by increased ovarian androgen production in quantity and quality by the increased number of follicular theca cells and then enhanced expression of steroidogenic enzymes in the theca cells [48]. Steroidogenic enzymes such as cytochrome P450 family members, including CYP11A (involved in cholesterol side-chain cleavage), CYP17A1 (also known as 17α-hydroxylase enzyme) [49], 3β-hydroxysteroid type II (HSD2B), and 20α-hydroxysteroid reductase (AKR1C1), are required for androgen production. A few studies have shown that the expression levels of those particular enzymes in ovarian theca cells are elevated in PCOS patients [50]. HA, which is caused by the theca cells found in PCOS women, exerts its hepatic impact through androgen receptors in the liver. The levels of SHBG, thyroxine- and cortisol-binding globulins, transferrin, and fibrinogen (hepatic synthetic products) decrease when exposed to androgens such as injectable testosterone formulation [51]. Hence, decreased SHBG levels may be considered one of the major clinical signs that the liver is unable to properly metabolize the androgen. Synergistically, insulin encourages ovarian androgen production in ovarian theca cells by activating the insulin-activated tyrosine phosphate cascade for increased glucose utilization [52]. As previously described, hyperinsulinemia is one of the key factors in androgen production. It starts a self-propagating positive feedback loop by lowering SHBG concentration and aggravating IR, exacerbating the HA or hyperinsulinemia symptoms over time.

Estrogen is considered a vital regulator in liver function and binds to estrogen receptors (ERs), which consist of two types: estrogen receptors α and β. The majority of the biological activities of estrogen in the liver are mediated through the estrogen receptor α, which contributes to the regulation of target gene transcription [53]; estrogen receptors are present in both animals and humans. However, the receptors only experience age-dependent decline rather than gender-based alterations in rat models [54]. The above-mentioned statements have shown a link between liver and sex steroid metabolism and their receptors. Phase I metabolism participates in androgen excretion in the presence of the CYP450/CYP3A group of enzymes, specifically CYP3A4, 5β reduction via 5β reductase AKR_1_D_1_, and oxidation via hydroxysteroid dehydrogenases [55]. In another study, the P-glycoprotein and testosterone elimination rates were much faster in women than men because of the enhanced concentrations of CYP3A4 [56].

### 2.5. Insulin Resistance

Insulin is a vital glucose and lipid metabolism regulator by direct or indirect action in the liver. Insulin receptors are heterotetramers composed of two extracellular α subunits and two transmembrane β subunits bound together by disulfide bonds. Once insulin binds to the α subunit of the insulin receptor, tyrosine residues in the β-subunit are autophosphorylated [57,58]. Insulin performs various functions in different insulin-dependent organs. For example, in skeletal muscle and the liver, insulin stimulates protein synthesis, glucose storage, and glycolysis. In the liver and adipose tissue, insulin initiates lipid synthesis and storage, while hepatic ketogenesis and gluconeogenesis in the liver are inhibited by insulin. It is well known that insulin concentration in the portal vein after food can be three times greater than in peripheral vessels [59]. When encountering this steeply increased insulin level, the liver activates insulin-dependent signaling pathways to accommodate the rise. Several mechanisms for this have been suggested, such as the translocation of glucokinase to the cytoplasm, the activation of hepatic glycogen synthases, and the reduced expression of enzymes involved in gluconeogenesis and glycogenolysis [60]. Activation of the insulin receptor kinase via insulin binding initiates the translocation of glucokinase (GCK)—an enzyme responsible for glucose phosphorylation leading to glucose retention within cells—from the nucleus to the cytoplasm to rapidly increase postprandial glucose uptake [61]. GCK is inactive while in the nucleus; however, it becomes activated after translocation from the nucleus to the cytoplasm and phosphorylates glucose to glucose-6-phosphate. This phosphorylation is the initial step of glycogen synthesis in the liver. The expression of GCK is subject to change at the transcriptional level when hyperglycemia is a long-term event. However, if the change in glucose level is relatively short-term, it may be resolved by the translocation of already-made GCK from the nucleus to the cytoplasm [61].

The tyrosine phosphorylation increases the intrinsic kinase activity of insulin receptors. It also involves the recruitment of intracellular signaling molecules (also known as adaptor proteins, such as insulin receptor substrates (IRS) and growth factor receptor-bound protein 2 (Grb2)) and effector proteins (e.g., phosphatidylinositol-3-kinase (PI3K) and mitogen-activated protein kinase (MAPK)). Downstream events after insulin receptor phosphorylation are categorized into two pathways: the metabolic PI3K pathway (initiated by the IRS and usually activated by the physiologic level of insulin) and the mitogenic MAPK pathway (initiated by Grb2 and Shc and usually activated at very high insulin levels) [59,62]. The metabolic pathway is mediated through the phosphorylation of IRS, the activation of PI3K, the inhibition of glycogen synthase kinase-3 (GSK3), and the membrane translocation of glucose transporter-4 (GLUT-4), which increases glycogen synthesis and glucose uptake. The mitogenic pathway is mediated through Grb2, Shc, and the activation of the RAS/MAPK cascade (MAPK is also known as extracellular signal-regulated kinases (ERKs)), resulting in the anabolic function of insulin. The insulin-like growth factor receptor 1 (IGF-1R) proteins are similar to the insulin receptor proteins, and both share nearly identical downstream pathways for the mitogenic actions of insulin [63].

Insulin, insulin-like growth factor 1 (IGF-1), and insulin-like growth factor 2 (IGF-2) interact with insulin receptors or with IGF-1R. Each ligand has a different affinity and potency for its receptors. In adipose tissue, a high level of insulin activates the MAPK pathways, promoting adipogenesis. In the liver, the importance of insulin-stimulated MAPK activation is not fully understood [64]. Hepatic glycogen synthases are activated by suppressing their inhibitor, glycogen synthase kinase, with insulin. 3-Phosphoinositide-dependent kinase-1 (PDK1) and mammalian target of rapamycin complex 2 (mTORC2) are intracellular mediators that are activated by insulin, ultimately leading to the initiation of hepatic glycogen synthesis and suppression of hepatic gluconeogenesis-sharing Akt—also known as protein kinase B, PKB—phosphorylation as an intermediate step [60]. Akt facilitates the translocation of GLUT-4, increases glucose uptake, and inhibits glycogen synthase kinase, promoting glycogen synthesis. The reduced expression of enzymes involved in gluconeogenesis and glycogenolysis is achieved by inhibiting forkhead box protein O1 (FoxO1), a transcription factor for the insulin-dependent expression of enzymes involved in hepatic gluconeogenesis and glycogenolysis. Insulin downregulates FoxO1, reducing the number of enzymes necessary for gluconeogenesis and glycogenolysis at the transcriptional level. FoxO1 is negatively regulated by Akt and MAPK, the effector proteins of insulin signaling pathways [65]. Phosphorylated FoxO1 by either Akt or MAPK is excluded from the nucleus, leading to an inhibition of FoxO1-dependent gene expression [66]. However, regulation at the transcriptional level is time-consuming and does not fully explain the acute changes in hepatic gluconeogenesis after facing postprandial hyperinsulinemia in the portal vein.

IR occurs when a large quantity of insulin is necessary to yield a normal response. In other words, insulin-dependent cells such as hepatocytes and skeletal and adipose cells are unable to respond effectively to normal physiologic insulin levels. Numerous studies have elucidated the underlying mechanisms of IR. The literature has identified components of insulin signaling pathways that contribute to insulin resistance if they are impaired, even though they do not effectively reveal the pathogenic molecules associated with IR. These components include the impairment of insulin synthesis in pancreatic β cells, accelerated insulin degeneration, loss of function of the insulin receptor, abnormality of an adaptor and effector proteins of intracellular insulin signaling, and mitochondrial dysfunction in target cells. One experimental study has shown that a rise in plasma free fatty acid (FFA) levels prompt insulin secretion in humans, which is sufficient to prevent an increase in plasma glucose levels [67]. However, this is only the case for acute rather than chronic IR. Triglycerides under chronic IR are also considered the primary culprit in IR [68].

IRSs are key adaptor proteins in an insulin signaling pathway. Four affiliates of the IRS family (IRS-1, IRS-2, IRS-3, and IRS-4) have so far been discovered. In humans, only IRS-1, IRS-2, and IRS-4 are present. IRS1 and IRS2 are located in the brain, fat, gill, heart, intestine, liver, kidney, spleen, white muscle, and ovaries and play a role in biological functions [69], whereas IRS-4 is located in the hypothalamus and thymus. Defects on the site of phosphorylation, for example, IRS-1 phosphorylation at serine 307, impair the insulin signaling pathways and cause IR. Diacylglycerol (DAG) is considered one of the molecular causes of IR in hepatocytes. It should be noted that lipids and triglycerides consist of three fatty acids and one glycerol. DAG is one of the intermediates in triglyceride synthesis. When exposed to a high-fat diet (HFD), DAG levels are increased in the hepatocytes. The accumulation of DAG promotes the translocation of protein kinase Cε (PKCε), and this kinase phosphorylates the insulin receptor at threonine 1160, which is located in the catalytic loop of the receptor. Threonine 1160 phosphorylation reduces the phosphorylation of the insulin receptor by destabilizing its active configuration, resulting in IR as a consequence [70]. The effect of DAG on PKC is short-lived, however, because the DAG-degrading enzymatic activities quickly remove the DAG. Interestingly, DAG is also formed in the endoplasmic reticulum (ER) during fatty acid esterification as an intermediate product. It is formed during triglyceride synthesis and stored in lipid droplet (LD) form [71]. DAG activates PKCε in hepatocytes and promotes the storage of activated PKCε, which may be strengthened by the association of the kinase, such as a receptor for activated C-kinase 2 (RACK2). RACK2 is known as the coatomer protein complex subunit β2 (COPB2). Coatomers are proteins located in membrane-bound transport vesicles that assist vesicle transport [71]. Two types of coatomers are currently known: coat protein I (COPI, retrograde transport of vesicles from the Golgi to the ER) and coat protein II (COPII, anterograde transport of vesicles from the ER to the Golgi). COPB2 is required for normal LD formation and is the main component of COPI vesicles, which are involved in the retrograde transport of cargo proteins from the Golgi to the ER, resulting in the placement/localization of PKCε to the Golgi.

To understand how insulin contributes to ovarian theca cell hyperproliferation and HA, the action of insulin in the ovaries is briefly summarized here. Studies have shown that insulin is mutualistically involved with FSH and encourages ovarian thecal-interstitial cell proliferation and differentiation. It also regulates folliculogenesis in the oocyte stage as well as egg maturation [72]. During fertilization, glucose absorption is essential for their maturation in response to insulin signaling [73]. In vitro investigations in cultivated polycystic ovarian theca cells revealed that insulin dramatically boosts progesterone and androstenedione synthesis compared to normal ones, and its activity is related to the direct actions of insulin on its receptors [74]. Thus, insulin acts as a co-gonadotropin that affects ovarian theca cells. In the multivariate analysis, obesity results in inflammation and IR, which further aid in the progression of PCOS [75], supporting the concept that androgen and insulin inhibit SHBG secretion and aggravate PCOS [76]. Another study has shown that stimulation of MAPK negatively regulates GLUT4, leading to impaired glucose transport and IR. Similarly, abnormal phosphorylation occurs at the serine part of the insulin receptor instead of the tyrosine part, resulting in selective impairment of insulin signaling and only a metabolic pathway, not a mitogenic pathway. This selectiveness is thought to be caused by the continuous stimulation of MEK1 and MEK2 (MEK1/2, members of MAPK kinase) by serine phosphorylated insulin receptors in PCOS [77,78]. Intriguingly, however, MEK inhibitors could not inhibit the stimulatory activity of insulin on 17α-hydroxylase activity, an enzyme required for androgen synthesis [79]. The results of a recent study revealed that miR-222 inhibition leads to decreased IR in PCOS, which is caused by stimulation of the PTEN and inhibition of the MAPK/ERK pathway [80]. The results of the above studies support the concept that IR and the accompanying hyperinsulinemia promote the pathogenesis of PCOS. Indeed, according to a recent study on Korean women with PCOS, approximately 20% of normoglycemic women with PCOS developed prediabetes or type 2 diabetes (T2D) after a median time of 2.9 years. This suggests that there is underlying IR in PCOS patients [81].

The result of a retrospective cohort study including 1545 women reveals that PCOS is an independent risk factor in women with gestational diabetes mellitus (GDM) for preeclampsia, obesity, elevated blood pressure, and blood glucose levels [82]. GDM is a common physiological condition among pregnant women whereby spontaneous hyperglycemia assists the demands of fetal growth. Insulin sensitivity reflects a metabolically responsive adaptation to gestational age. During the first trimester of pregnancy, insulin sensitivity increases to promote the uptake of glucose into adipose tissue, preparing for the energy demands of the second and third trimesters [83]. However, as pregnancy progresses, IR is increasingly promoted by placental hormones and cytokines, including estrogen, progesterone, leptin, cortisol, placental lactogen, and placental growth hormone [84]. As the barrier between the mother and fetus, the placenta itself is also affected by hyperglycemic physiologic environments during GDM. This pathophysiologic environment exposed to the GDM may not only affect the transport of glucose, amino acids, and lipids across the placenta but also produce endogenous glucose and facilitate the breakdown of fat. Thus, GDM may result from the increase in blood glucose and FFA concentrations through a feed-forward pathway [85].

It is also known that insulin sensitivity decreases by about 30–60% at full term compared with before pregnancy [86]. In the context of glucose homeostasis, pregnant women compensate via hypertrophy or hyperplasia of pancreatic β-cells and increased secretion of glucose-stimulated insulin [87]. Since placental hormones promote IR, we can presume that insulin sensitivity rapidly returns to pre-pregnancy conditions within a few days post-partum with the placental release [88]. Although GDM usually resolves itself following delivery, it may also cause health problems for both mother and fetus, such as overt DM, obesity, and other metabolic diseases.

The liver encounters the dietary fats originated by adipocytes via lipolysis and increased de novo lipogenesis of the hepatocytes. The intravenous provision of lipids for less than two hours increases intramyocellular glucose-6-phosphate concentrations and reduces glucose utilization. Previous studies have demonstrated that enhanced fatty acid oxidation increases the ratio of intra-mitochondrial acetyl coenzyme A (CoA) to CoA and NADH to NAD^+^ [89]. The increased phosphofructokinase (PFK) inhibition is a crucial rate-controlling enzyme in glycolysis. Hence, lipid-induced defects in glucose transport caused by IR affect insulin sensitivity in muscle but do not affect a lipid-induced reduction in pyruvate dehydrogenase activity [90].

Insulin directly enhances sterol regulatory element-binding protein-1 (SREBP-1) expression by modulating mTOR activity, resulting in IR [91]. Another study’s results show that the inhibition of SREBP-1c attenuates hepatic IR by altering hepatic lipid metabolism through the interference of the mTOR-dependent pathways [92]. Hence, the SREBP-1c upstreaming depends on the mTORC1 induction, and mTORC1 is independent of the insulin signaling in liver insulin receptor knockout mice [92]. These results indicate that inhibition of SREBP-1c attenuates hepatic IR by altering hepatic lipid metabolism via interfering with the mTOR-dependent pathways. SREBP-1 also plays a role in controlling the activity of L-pyruvate kinase, acetyl-CoA carboxylase, fatty acid synthase, stearoyl-CoA desaturase 1, mitochondrial glycerol-3-phosphate acyltransferase 1, and other genes encoding for lipogenesis and glycolysis pathways.

Liver-compromised diseases such as NAFLD patients were reported to have elevated FFAs in comparison with the healthy group, which is a sign of increased insulin levels and its associated reduced lipolysis with resultant significantly higher FFAs in the bloodstream [93]. Recently, a genome-wide association study (GWAS) was conducted on ten population-based cohorts (14,751 individuals, 56% women) to see metabolic risk factors that interact with the genetic risk of NAFLD. The study reported a strong association between IR and patatin-like phospholipase domain-containing protein 3 (PNPLA3) in the progression of hepatic fibrosis. PNPLA3 was also discovered by another GWAS in a population comprising Hispanics, African Americans, and White Americans [94]. According to the study, the PNPLA3 allele was strongly associated with increased hepatic fat levels and inflammation. PNPLA3 homozygotes have a hepatic fat content that is more than twice the normal level.

Apart from PNPLA3, there are several other gene products, including uridine diphosphate (UDP)-glucose ceramide glucosyltransferase (UGCG)—an enzyme involved in the synthesis of ceramide, a component of the cell membrane—and mitogenic protein kinase 4, which are directly involved in insulin signaling and are related to liver fat accumulation. A study conducted by Greco et al. investigated gene expression levels using liver biopsy samples from NAFLD patients with severe steatosis and healthy controls. Genes such as PLIN, ACADM (both lipid metabolic genes), FABP4, CD36 (both genes are involved in fatty acid transport), BCAT1 (an amino acid catabolic gene), and CCL2 (involved in inflammation) are upregulated in the fatty liver [95].

Another gene, SLC16A13, which was once identified as a novel susceptibility gene for T2D, is also associated with NAFLD. Slc16a13 knockout mice were found to attenuate HFD-induced lipid accumulation in the liver and IR. Therefore, the study authors suggested that SLC16A13 could be a potential target for treating NAFLD and T2D [96]. Notably, a recent study has shown that a gna13 (Gα13) deficit in hepatocytes in response to hyperglycemic stimuli causes hyperglycemia and IR in other organs, such as the skeletal muscles and adipose tissue [32]. G proteins belong to the G protein family and act as molecular switches when GPCR is activated by stimulatory ligands [97]. G proteins consist of three subunits: α, β and γ. The α subunit of the G protein is the key part that functions as a molecular switch. Therefore, Gα proteins are key modulators for physiological and biochemical activities [98]. Liver-specific deletion of gna13 (Gα13 LKO) showed no significant difference in liver steatosis compared to the wild-type but did show a higher fasting glucose level and systemic IR than the wild-type. The induction of O-GlcNAc transferase results in hypersecretion of ITIH1 from the liver, which is then deposited in the hyaluronan surrounding adipose tissue and skeletal muscle, making insulin less likely to interact with the insulin receptors in those tissues [32]. O-GlcNAcylation is thought to affect protein stability by modifying protein ubiquitination [99]. Given that ITIH1 overproduction causes IR in adipose tissue and skeletal muscle. It is also possible that ITIH1 overproduction may also lead to IR in ovarian cells, which could contribute to the development of PCOS. However, further research is needed to determine whether the potential deposition of ITIH1 onto the surface of ovarian follicles is responsible for IR (as shown in Figure 2). This hypothesis is further supported by the finding that ITIH and its subtypes, hyaluronic acid-binding proteins, exist in the ovary around theca and stomal cells surrounding the mature follicles and follicular fluid, and on the cumulus cell surface during ovulation [100]. The above idea may also explain the transition of GDM to overt DM after delivery although this also needs to be investigated further.

Other conditions related to IR are briefly listed here. The relationship of obesity with gut-associated IgA levels explains why IgA concentration decreased drastically, resulting in the aggravation of IR in mice [101]. A study published in 2018 shows that zinc transporter-7 (ZnT-7) is present in skeletal muscle, adipose tissue, β-cells, and the small intestine, while the knockout mice of ZnT-7 result in intracellular lipid deposits, which also cause increased oxidative stress and inflammation via inflammatory mediators as well as IR [102].

### 2.6. Oxidative Stress

Lipid accumulation in hepatocytes causes lipotoxicity by reducing oxidative phosphorylation and altering mitochondrial calcium influx, resulting in electron transport chain dysfunction, which also produces ROS. Mitochondria are the primary regulators in maintaining balance among the β-oxidation of FFA, electron transport chains, ATP production, and ROS. All the oxidized reactions occurring in mitochondria (e.g., the tricarboxylic acid cycle) are interlinked with the conversion of NAD^+^ and FAD^+^ into their reduced forms, NADH and FADH_2_, respectively [103]. The resulting electrons from this reaction are channeled via respiratory chain carrier complexes I, III, and IV and end at the electron acceptor activity of molecular oxygen [104,105]. The abnormal activities of mitochondria cause an imbalance between pro-oxidant and antioxidant activities, which further leads to the reduced metabolism of fatty acids and subsequent ROS production [106]. Oxidative stress interrupts signaling and redox balance, further exacerbating ROS activity, which damages the mitochondrial membrane and compromises the activity of the antioxidant defense system [106].

At the molecular level, ROS causes tyrosine nitration, mitochondrial DNA damage, and membrane depolarization [107]. In addition, ROS production is closely linked to the activation of JNK and NF-κB, which may lead to apoptosis [108]. Commonly, upon exposure to oxidative stress, antioxidants such as superoxide dismutase (SOD) and glutathione peroxidase (GPx) are activated to facilitate ROS detoxification [109]. Various research studies have proven that the levels of antioxidants such as glutathione, catalase, and SOD are significantly lower in animals with non-alcoholic fatty liver disease (NAFLD). Treatment with antioxidants subsequently restored or improved glutathione, catalase, GPx, peroxiredoxins, and SOD levels in NAFLD patients [110]. Results of another study have shown that serum glutathione levels were significantly reduced in the serum of NAFLD patients, while there were enhanced levels of ALT and the DNA oxidation marker 8-hydroxy-2-deoxyguanosine (8-OHdG). However, the levels of these markers were significantly decreased after antioxidant therapy [111].

PCOS pathogenesis has also been directly linked to oxidative stress. Up until now, with cutting-edge research, it has been proven that ROS-induced damage may be retained in the ovaries, where it can diminish oocyte quality and also cause the apoptosis of granulosa cells, resulting in anovulation and degeneration of the corpus luteum [112,113]. In another study, PCOS patients showed enhanced mononuclear cell-associated ROS production and p47phox expression in response to elevated glucose levels. This surge in ROS can induce a pro-inflammatory cascade and cause HA [114]. Therefore, it appears that elevated oxidative stress is primarily responsible for anovulation, enhanced glucose levels, and HA [114].

Studies have proven that skeletal muscles are considered the primitive site for insulin-mediated glucose clearance. IR in the skeletal muscle is considered a major risk factor in PCOS patients, as well as diabetes. In the experimental approaches, the compromised insulin-to-glucose response via Akt accounted for the GLUT4 content in cultured skeletal muscle tissue [115]. The overexpression of miR-93 in PCOS and control subjects with IR was found to be elevated and involved in suppressing GLUT4 expression [116]. Similarly, augmented miR-223 in adipose tissue positively regulates the HOMA-IR by reducing GLUT4 protein and insulin-mediated glucose uptake [117]. It also plays a role in inflammation, energy stress, and other physiological events. The loss of the miR-33 gene might result in pronounced IR and increased lipid content in the livers of mice, even without HFD feeding [118]. The ablation of muscular Gα13 positively affects skeletal muscle-induced fiber-type switching and IR [119].

### 2.7. Inflammasomes

Cell death is one of the major regulators in maintaining the cell cycle, as it removes damaged cells from the body [120]. Janeway explained innate immunity and proved that antigen-presenting cells that are expressed on receptors accessible to microbial products further exacerbate the inflammatory response [121]. Lipotoxicity, which is caused by the activation and release of danger-activated molecular patterns (DAMPs) from damaged hepatocytes as well as the gut microbiome, facilitates the release of pathogen-activated molecular patterns (PAMPs) [122]. TLR4 on the Kupffer cell (KC) membrane stimulates the production and release of the cytokines responsible for inflammation. TLR4 stimulation triggers activation of p38, MAPK, PI3K, and NF-κB, followed by enhanced cytokine production [123]. Translocation of PAMPs occurs via portal blood into the liver and binds to the pattern-recognition receptors on hepatic immune cells, including Toll-like receptors (TLR), initiating an inflammatory response [123]. The inflammasome is a protein complex that can detect danger signals through nucleotide-binding oligomerization domain receptors called nucleotide-binding oligomerization domain (NOD)-like receptors (NLRs). Four types of inflammasomes have been discovered: NLR family pyrin domain-containing 1 and 3 (NLRP1, 3), NLR family CARD-containing protein 4 (NLRC4), and melanoma-2 (AIM2). A series of microbes could activate NLRP3 and host-derived triggers, including bacteria, viruses, DAMPs, and pore-forming toxins [124]. In particular, NLRP3 is one of the most significant contributors to the development of nonalcoholic steatohepatitis (NASH) and liver fibrosis in mice [125]. NLRP3 is activated in two ways: canonical and non-canonical. The canonical pathway is activated by transcription and oligomerization. In the context of NAFLD, macrophages, including KC, are the initial central hub for cytokines and chemokines, including TNFα, IL-1β, and CCL2. Microbial [126] or endogenous cytokine production results in the stimulation of NLRP3 and pro-interleukin-1β via NF-κB.

Caspase-8 and FAS-associated death domain proteins (FADD) act as primers by regulating the NF-κB pathway. Lys-63-specific deubiquitinase BRCC36 (also known as BRCC3) and IL-1 receptor-associated kinase 1 (IRAK1) play a role in the activation of the NLRP3 inflammasome [127]. Caspase recruitment domain (CARD) by an adaptor protein called ASC (apoptosis-associated speck-like protein containing a CARD) stimulates cysteine protease procaspase-1, leading to the maturation and release of pro-inflammatory cytokines and programmed inflammatory cell death [128]. Viral RNA stimulates the NLRP3 inflammasome. Potassium efflux is also one of the major stimuli of NLRP3 activation, while Ca^2+^ signaling causes ER stress and mitochondrial dysfunction, as well as NLRP3 inflammasome activation [127]. In addition, NLRP3 activation has also been shown to cause ovarian fibrosis in mice [129].

In PCOS, miR-1224-5p deactivates the NLRP3 inflammasome via FoxO1 [130]. The ability of granulosa cells to produce mature oocytes is arrested because of inflammatory reactions, which have also been confirmed in patients with PCOS [131]. The WNT5a, which is a signaling cascade, has recently been associated with inflammatory conditions such as obesity, psoriasis vulgaris [132], sepsis, endothelial inflammation [133], liver fibrosis, and cirrhosis [134]. It has also been shown that WNT5a expression activates inflammatory genes by specifically upregulating the PI3K/AKT/NF-κB signaling pathway, further leading to PCOS aggravation [135]. The transformation of growth factors such as AMH, inhibins, activins, bone morphogenetic proteins [136], growth differentiation factor 8 [137], and other cytokines promotes the irregular-shaped follicle development found in PCOS [138].

### 2.8. NAFLD

In the case of NAFLD, the compromised liver leads to decreased androgen metabolism and increased metabolism of SHBG, which propagate through different pathways and cause HA, hyperinsulinemia, and resultant infertility [139]. Moreover, many studies have reported increased ALT and AST activities cumulatively in PCOS patients [140]. Previous experimental studies and clinical profiling of patients have proven that variations in BMI are directly associated with plasma SHBG levels in men and women [141]. Reduced SHBG and IGF, preferentially insulin-like growth factor-binding protein 1 (IGFBP-1, also known as placental protein 12 or PP12), have been reported to be associated with hyperinsulinemia and represented as critical biomarkers/indicators of the metabolic syndrome [142].

The results of the meta-analysis comprising compromised hepatic diseases reveal that SHBG levels significantly decreased in women, whereas serum total testosterone levels increased [143]. Studies have proven that reduced SHBG levels, HA, hyperinsulinemia, and T2D are the primary symptoms of PCOS [144]. Thus, increased levels of testosterone are indicative of PCOS and may also be associated with NAFLD. The results of the meta-analysis and systemic review of 17 studies comprising 2734 PCOS patients and 2561 healthy and BMI-matched women reveal that PCOS women who also have NAFLD exhibited increased serum testosterone levels and an increased free androgen index compared with those who did not have NAFLD [11]. A retrospective longitudinal cohort study was performed in the United Kingdom using the primary care database, which included 63,120 NAFLD women suffering from PCOS and 121,064 healthy and BMI-matched women from January 2000 to May 2016. The results showed that women with serum testosterone >3.0 nmol/L were 2.3 times more likely to be diagnosed with NAFLD than those who had normal testosterone levels (hazard ratio (HR) = 2.30, 95% confidence interval (CI) 1.16–4.53, *p* = 0.017). Additionally, women with SHBG < 30 nmol/L were 4.75 times more likely to be diagnosed with NAFLD than those with normal SHGB levels (HR = 4.75, 95% CI 2.44–9.25, *p* < 0.001) [13]. A case-control study including 29 young PCOS patients and 22 healthy controls used proton magnetic resonance spectroscopy to confirm that women with PCOS have increased levels of hepatic fat compared to control individuals, confirming that PCOS patients have a risk of NAFLD [145]. Another case-control study, including 275 serum samples of PCOS patients selected under the Rotterdam criteria, was analyzed by liquid chromatography–mass spectrometry (LC–MS). The result showed that PCOS patients had increased testosterone, dihydrotestosterone, fasting glucose, ALT, and AST levels, which are clearly indicative of hepatic disease [146].

### 2.9. Genetic and Epigenetic Factors

PCOS has a substantial hereditary component [147]. It is a familial disorder with a single autosomal prevailing gene impact and variable phenotype, and family history is considered one of the main risk factors for developing PCOS [148]. Data obtained from another clinical study has shown that women have a ~40% chance of developing PCOS if their sister is also affected [149]. The univariate genetic model results show that there is a 66% genetic variance in PCOS cases [150]. The risk of daughters developing PCOS if their mothers have it increases five-fold [151]. Thus, PCOS is a heterogeneous disorder that has a strong genetic component the influence.

The results of the oocyte-specific proteome identified 46 proteins that are secreted during the oocyte stage, which include extracellular matrix proteins such as ZP1–4, ACAN, KAL1, proteases (e.g., ACE, OVCH, ASTL, HTRA1), SERPINs, and ITIH1 [152]. In the “post-GWAS” era, the aims of studies should be directed toward elucidating the relationship between genotype-phenotype traits and the role of environmental changes. Genome-wide association analysis (GWAS) of women suffering from PCOS has found more than 15 susceptible gene loci. Subsequently, several genes, such as INSR (genes for insulin receptor), FSHR (genes for FSH receptor), C9orf3, DENND1A, THADA, ARL14EP, and GATA4/NEIL2, have been validated [153,154]. More interestingly, according to Mendelian randomization analyses, the risk factors for PCOS include lower levels of SHBG, increased BMI, and IR [155]. Another study shows that a reduced level of SHBG is one of the primary causes of PCOS progression [156]. A further recent GWAS data analysis revealed that rs12478601, rs2059807, rs4784165, and rs2479106 were found on the introns of THADA, INSR, TOX3, and DENND1A, respectively, suggesting that they may be involved in the progression of PCOS in association with IR and other metabolic disorders [157].

Non-coding RNA involved in ovarian functions was also downregulated in PCOS patients because of altered methylation [158], which implies that epigenetic factors caused by environmental changes also have an impact. In women with PCOS, the researchers also discovered reduced DNA methylation in the LHCGR (gene for LH/choriogonadotropin receptor) locus and increased methylation in the INSR locus. Interestingly, the DNA methylation procedure in subcutaneous adipose tissue exhibits plasticity, which may be reversed by weight loss or the use of treatments [159]. In addition, 3840 genes were found to be involved in Wnt signaling, G protein receptors, and endothelin/integrin chemokine/cytokine-mediated inflammation, exhibiting variations in methylation in PCOS patients. Mimouni et al. identified six genes that are hypomethylated in PCOS patients, including TIT1 (involved in DNA methylation), ROBO-1 (axon guidance), CDKN1A (inhibition of cell proliferation), two insulin signaling-associated genes (IGFBPL1 and IRS4) and inflammation (HDC) [160].

## 3. Medications and New Drugs under Development

### 3.1. Medications

The principal intervention for metabolic dysfunction in PCOS is a lifestyle modification, which improves fertility. According to the studies, a lifestyle modification enhances ovulation in PCOS patients, and women have spontaneous pregnancies. Physical exercise can reduce IR, total fat content, and cardiovascular events [161]. Weight loss in obese PCOS women positively boosts metabolic profiles and induces ovulatory cycles [162]. Physical exercise has reduced IR, total fat contents, and cardiovascular events [161]. However, when lifestyle modifications are not sufficient to improve PCOS symptoms, pharmacological intervention may be necessary (Table 1).

If anovulatory PCOS patients want to become pregnant, ovulation can be induced using oral drugs such as clomiphene citrate or letrozole. Since the hypothalamus is the center of negative feedback from circulating estrogen levels, clomiphene prevents the hypothalamus from sensing the circulating estrogen levels and thereby increasing GnRH secretion. Clomiphene citrate is categorized as a selective estrogen receptor modulator (SERM) and competes with estrogen at estrogen receptors in the hypothalamus [163]. Clomiphene binds to estrogen receptors for longer periods than estrogen, making the estrogen receptors depleted in the hypothalamus [163]. Clomiphene citrate enhances the chance of pregnancy compared with the placebo group, while it could decrease the live birth or ongoing pregnancy ratio when compared with gonadotropin [198]. Letrozole, which is a member of the aromatase inhibitors, may also induce ovulation. By inhibiting aromatization with letrozole, estrogen production is directly reduced, leading to a compensatory increase in pituitary gonadotropin by a negative estrogen feedback. Letrozole is the current drug of choice for inducing ovulation in PCOS patients [175]. The results of a meta-analysis including 3962 women with PCOS show that letrozole significantly improves live birth and clinical pregnancy rates and decreases time-to-pregnancy compared to clomiphene citrate [199]. However, when neither clomiphene nor letrozole can induce ovulation, a gonadotropin injection is often administered. In this case, low doses of gonadotropin, between 75 and 150 IU/day, have been effective in PCOS patients and have reduced the incidence of ovarian hyperstimulation, a common side effect in women with PCOS using gonadotropins [200].

Similar to PCOS, NAFLD can usually be managed by lifestyle modifications such as exercise or following a low-calorie diet [201]. However, if the condition worsens, the disease progresses to NASH, or the lifestyle modifications are ineffective, then pharmacological intervention will be necessary. The result of a meta-analysis shows that acarbose treatment in PCOS patients reduces testosterone, triglycerides, and very low-density lipoprotein (VLDL) and increases high-density lipoprotein (HDL) levels [166]. The effects of statins in combination with metformin in PCOS patients showed decreased C-reactive protein (CRP), triglycerides, total cholesterol, and low-density lipoprotein (LDL) levels [168]. In women with PCOS who also exhibit hyperglycemia and IR, metformin is the drug of choice as it can reduce obesity [202]. The reduced insulin levels as a result of metformin therapy lead to decreased androgen production, thereby regulating the menstrual cycle [167,203]. The results of the meta-analysis revealed that metformin administration leads to reduced fasting glucose levels in PCOS [204]. Thiazolidinediones are considered an alternative treatment for NAFLD-associated PCOS. The drugs interact with nuclear receptors (i.e., PPAR, peroxisome proliferator-activated receptors) and act as an insulin sensitizer, preferably in adipose tissue and skeletal muscle, also improving BMI and triglyceride levels [170]. In another study, 137 PCOS patients treated with both myo-inositol and D-chiro inositol for six months were shown to significantly improve their menstrual cycle, IR, and acne score [172]. The findings of another study suggest that supplementation with Myo-Inositol (1200 mg per day) and D-Chiro-Inositol (135 mg per day) for 12 months significantly improved FSH, LH, and progesterone as well as oligomenorrhea and follicle count [205].

An Endocrine Society Clinical Practice Guideline states that oral contraceptives are considered first-line therapy in managing menstrual problems and hirsutism/acne of PCOS [173]. If oral contraceptives are ineffective, anti-androgen drugs such as spironolactone (50–100 mg twice daily) and finasteride (5 mg daily) can be used for hirsutism. The hormonal treatment with low doses of FSH and LH has been found effective in PCOS patients by reducing the incidence of ovarian hyperstimulation, which is considered one of the significant risk factors in PCOS [165]. Eflornithine, a reversible inhibitor of ornithine decarboxylase (required for cell division), is effective in halting unwanted hair growth if applied to skin areas [1,175].

Currently, however, there are no Food and Drug Administration (FDA)-approved therapies for NAFLD [169,201]. Ursodeoxycholic acid, omega-3 fatty acids, and metformin, despite the initial promise, failed to show a lasting histological benefit [169]. Only pioglitazone and vitamin E have shown histological benefits [176,206]. Pioglitazone, which acts as a PPARγ agonist, enhances adipocyte fat storage and thereby relieves NASH symptoms [176]. Vitamin E, also known as tocopherol, acts as an antioxidant, reducing oxidative stress and possibly regulating apoptosis [207]. However, vitamin E has potential adverse effects, such as an increased risk of bleeding and cardiovascular disease [169]; hence, vitamin E usage is only limited to cases of biopsy-proven NASH without diabetes [208]. Vitamin D (cholecalciferol) deficiency (serum 25-hydroxyvitamin D < 30 ng/mL) has been considered an aggravating factor of IR in PCOS patients. Therefore, vitamin D supplements are occasionally prescribed to PCOS patients [209], as they can significantly reduce testosterone levels, reduce hirsutism scores, and increase glutathione contents [209]. Interestingly, vitamin D deficiency is also associated with NAFLD development [210]. Several clinical trials have been designed to verify the clinical efficacy of vitamin D supplements on NAFLD [211,212], but have given conflicting results thus far. Additionally, a recent meta-analysis of 467 patients from nine trials failed to show any clinical efficacy of vitamin D supplements on NAFLD treatment in terms of serum aminotransferase levels [213]. Therefore, vitamin D supplements are not currently included in the clinical management guidelines for PCOS [175,214] and NAFLD [206].

### 3.2. New Drugs under Development

As the available data and research activity for the cure of PCOS patients is limited, there is a need for the best alternative therapy that should be economical and easily accessible with reduced side effects and adverse drug reactions. Studies have proven that there is a significant association between PCOS and endometriosis [215], while another study’s results show that 76% of patients with endometriosis in stages I and II were reported to have PCOS [216]. Elagolix (ORILISSA™) is a second-generation, low-molecular-weight non-peptide GnRH receptor antagonist that lowers blood levels of ovarian sex hormones. The FDA approved Elagolix 150 mg and 200 mg tablets in 2018 to treat moderate to severe pain associated with endometriosis [177].

While supporting our hypothesis and the scientific evidence that liver diseases, specifically NAFLD, play a significant role in the progression of PCOS, we would like to recommend some drugs that are currently in the pipeline for NAFLD treatment. These drugs have the potential to effectively manage and treat PCOS by reducing its symptoms and improving liver function in affected patients. Because of the compromised liver, there might be no proper cure for PCOS patients except for its management. Studies have proven that due to the compromised liver in NAFLD, there is reduced SHBG, hyperinsulinemia, elevated ALT and AST activities, and testosterone, which must be managed to cure PCOS. Therefore, according to the above-stated reasons, we suggest that drugs currently in the pipeline can be used as alternative treatment options for managing PCOS symptoms. It is a glimpse of the curated Biomedtracker database’s phases I–III and new drug applications or biological license applications for potential therapeutic agents in the NASH domain [179]. The NASH pipeline is active, with four candidates in phase III. Preclinical, investigator-initiated, and investigational new drug candidates are not part of this category. Phase I/II and phase II/III are counted separately as phase II and phase III, respectively.

Resmetirom is a selective thyroid hormone receptor-β (THRβ) agonist. A phase IIb study enrolled 348 patients, out of whom 233 were excluded, i.e., not meeting the criteria (*n* = 181), withdrew (*n* = 28), lost follow-up (*n* = 12), physician decision (*n* = 1), and random reason (*n* = 1), while 125 patients were included. A total of 84 patients were randomly assigned to resmetirom and 41 to placebo. As per MRI-PDFF measurements, patients treated with resmetirom for 12 and 36 weeks showed a reduction in hepatic fat (32.9 and 37.3%, respectively) compared to those who took a placebo at the same time (10.4 and 8.5%, respectively). Liver biopsy results of resmetirom showed 27% resolution in NASH [178]. Phase III study results show that resmetirom at the dose of 80–100 mg/day for 52 weeks was found to be effective in the treatment of NASH [179]. VK2809 is another thyroid hormone receptor β agonist for liver tissue and has been found effective against various hyperlipidemic ailments, such as NASH. It stimulates genetic transcription and upregulates mitochondrial fatty acid oxidation, resulting in reduced lipid blood serum levels. A study has been performed by Viking Therapeutics’ phase IIb. It was randomized, double-blind, placebo-controlled, and multiple areas were selected for the study, including patients with F1-F3 stage corroborated by biopsy among 337 patients at the dose of 1 and 2.5 mg daily, 5 and 10 mg every other day, and one placebo group. After 12 weeks, the drug showed a reduction in fat content; however, after 52 weeks, liver fibrosis decreased [217]. It is supposed that it may be effective in improving the lipid profile of PCOS patients by improving hepatic lipid content.

AMPK acts as a sensor and regulates metabolic and inflammatory issues. PXL770 is an agonist of AMPK, and its efficacy is studied against non-alcoholic steatohepatitis and adrenoleukodystrophy. The clinical study results revealed that it inhibits liver lipogenesis and enhances glucose metabolism and insulin sensitivity in NAFLD patients [180]. Because of improved blood glucose levels and IR, the candidate may be found to be effective against PCOS.

Tirzepatide is a glucagon-like peptide-1 (GLP-1) receptor stimulant and activator that synergizes the action of incretins into a single novel molecule [181]. In the clinical data provided by Eli Lilly and Company, experimentally, Tirzepatide was given to NASH patients at doses of 5, 10, and 15 mg subcutaneously per week; Trizepatide treatment for 52 weeks showed a relative reduction in hepatic lipid content [182]. The GLP-1/GLP-1R axis stimulates the granulosa cells by partially altering the FoxO1 phosphorylation and synergizes the oocyte maturation [218]. Because of the effect mentioned above, GLP-1 agonists are considered therapeutic agents for managing PCOS symptoms, such as diabetes and IR. Cotadutide, a GLP-1 agonist, has alleviated fibrosis more significantly than liraglutide or obeticholic acid and exhibits weight loss in diabetes in two preclinical mouse models of NASH [185]. It has also exhibited promising results in a phase I trial in Japanese patients [184]. Moreover, it may be a potential treatment option for PCOS.

Aramchol (phase IIb drug) is an adjoined cholic acid and arachidic acid and enables reducing liver steatosis by preventing the stearoyl-CoA desaturase 1 (SCD1), an enzyme involved in controlling the rate-limiting step in the mono-unsaturated fatty acid synthesis. A study was conducted on 60 patients with NAFLD who were treated with aramchol at doses of 100 and 300 mg once/day for 90 days and showed a dose-dependent reduction in liver fat content [186]. Another study, which included 247 patients, exhibited a reduction in NASH and improved fat ALT, AST, and HbA1c levels at doses of 400 and 600 mg for 52 weeks. Moreover, the results of the 600 mg treatment have shown a pronounced effect on NASH [187].

Belapectin is a carbohydrate complex known as galactoarabino-rhamnogalacturonate (GRMD-02), which is derived from natural plants. It is a galectin inhibitor that interacts explicitly with galectin associated with the membrane and secreted from the membrane and has also been reported to be effective in treating liver fibrosis and hypertension [188,189]. A study conducted in phase IIb included 162 patients with NASH, liver cirrhosis, and hepatic venous pressure gradient who received the belapectin treatment twice a week for one year. However, the results showed a non-significant reduction in hepatic venous pressure gradient (HVPG) or fibrosis compared with a placebo. In another group of patients without esophageal varices, 2 mg belapectin did reduce HVPG and the development of varices [188].

Efruxifermin, an FGF21 analog (phase IIa), was evaluated with once-weekly dosing, including 80 patients with biopsy-proven NASH (NCT03976401) who were given efruxifermin at three different dose levels (28 mg, 50 mg, and 70 mg) for 16 weeks and showed a significant reduction in hepatic fat in patients with NASH and F1–F3 fibrosis. Pegbelfermin is a pegylated analog of FGF21 and was used for NASH patients during the phase IIa trial, where it showed a meaningful reduction in hepatic fat percentage at the dose of 10 mg/day. However, 16% of patients experienced the adverse effects of diarrhea [191].

The lanifibranor (indole sulfonamide), also known as IVA337, is a PPAR agonist and showed antifibrotic activity in the db/db mouse model of CCl_4_-induced liver fibrosis [219]. In two hundred forty-seven patients (42% with type 2 DM and 76% with moderate or advanced fibrosis), lanifibranor showed 49% protection against steatosis activity and fibrosis compared to the placebo, which exhibited 22%. It has also significantly decreased the majority of inflammatory and fibrotic biomarkers; however, patients have experienced adverse effects such as peripheral edema and anemia [192].

Another emerging drug, aldafermin, was found to be effective against NASH during phase II by reducing the liver lipid and improving its functions [220], and it may be found effective against PCOS because improving the hepatic function will also be of help for hyperandrogenism issues in women.

CER-209, an agonist of the G protein-coupled P2Y13 and currently in phase I, has been reported to decrease steatohepatitis, which was attributed to the reduction in cholesterol and triglycerides in comparison with a placebo group. Furthermore, it has provided new insight into the basis for curing NASH, as it is being stated by the company that it acts through the reverse lipid transport pathway [196]. It is also proven that P2Y levels decreased in experimentally induced PCOS rats compared to the control group, as normal levels are required for the proliferation of theca cells [197]. We can hypothesize that CER-209 may be effective for patients with PCOS.

## 4. Concluding Remarks

Most of the symptoms associated with PCOS are linked to IR, T2D, decreased SHBG levels, and increased ALT/AST levels. The body’s fuel metabolism and ovarian function can be impacted by any pathological changes at the levels of systemic metabolism and/or in peripheral organs, particularly the liver, which serves as the metabolic center of the body, potentially through the liver-to-ovary axis. This review aims to elucidate the relationship and underlying mechanistic basis between PCOS and metabolic disorders by archiving the common pathological features, the associated molecules, and pathways, and providing recent information on the treatment strategies.

## Figures and Tables

**Figure 1 ijms-24-07454-f001:**
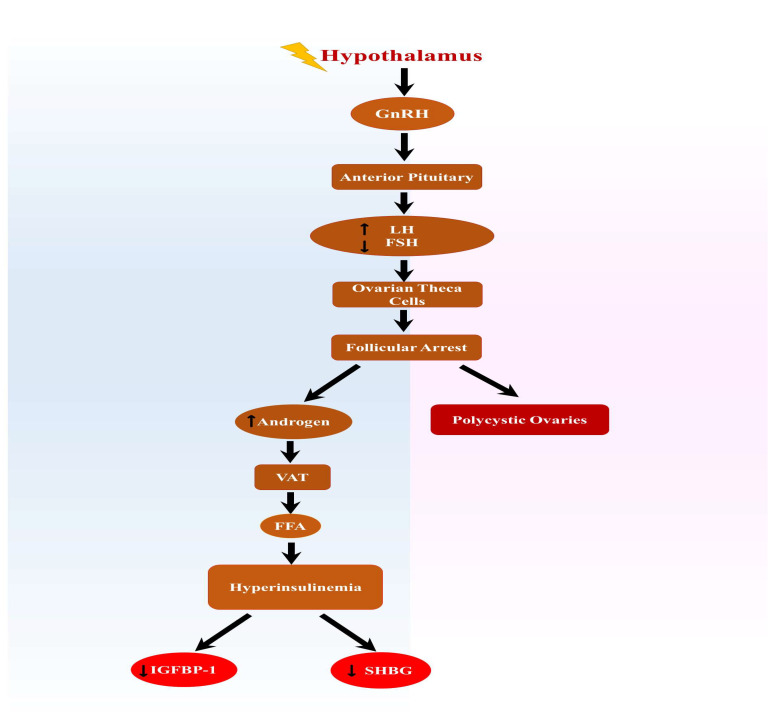
Overstimulation of the hypothalamus results in the enhanced secretion of luteinizing hormone and decreases FSH. A decrease in FSH prevents follicular development and enhances androgen release, leading to polycystic ovaries. Elevated androgen levels stimulate VAT, which increases FFA and results in hyperinsulinemia with reduced SHBG and IGFB-P1. Abbreviations: FSH, follicular stimulating hormone; VAT, visceral adipose tissue; FFA, free fatty acids; SHBG, sex hormone binding globulin; and IGFBP-1, insulin-like growth factor-binding protein 1.

**Figure 2 ijms-24-07454-f002:**
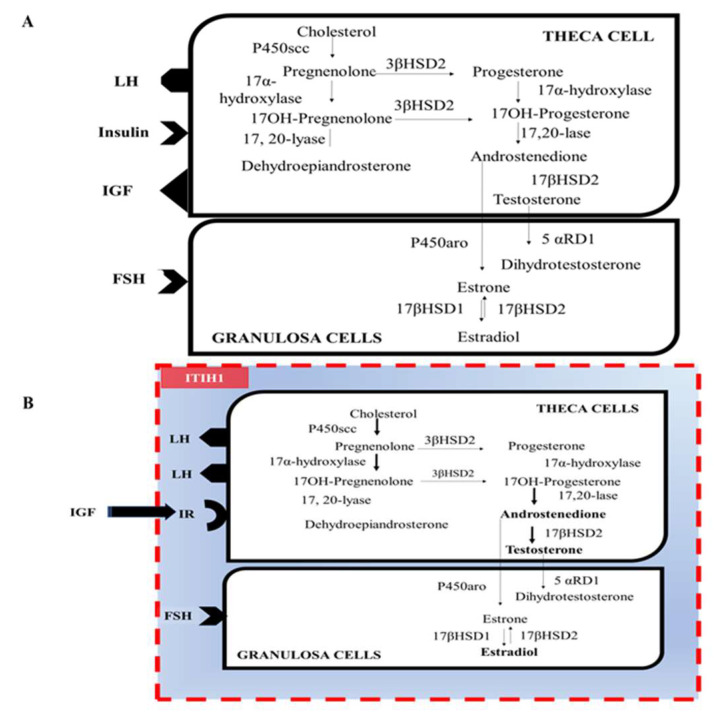
Binding hormonal ligands to the receptors in ovarian cells and hypothetical ITIH1 interference. (**A**) The bindings of LH, insulin, IGF, and FSH to the cell surface membrane-bound receptors in ovarian cells. (**B**) The potential effect of ITIH1 on ovarian cells. Reduced levels of Gα13 in the liver lead to the hypersecretion of ITIH1 from the liver. Circulating ITIH1 may travel to the ovarian theca and granulosa cells. The binding of ITIH1 to HA on the cell surface may prevent insulin from binding to its receptors. Abbreviations: LH, luteinizing hormone; IGF, insulin-like growth factor; FSH, follicular stimulating hormone; ITIH1, inter-alpha-trypsin inhibitor heavy chain H1; and HA, hyaluronic acid.

**Table 1 ijms-24-07454-t001:** Medications and drug candidates under development for PCOS treatment.

**Drugs Available for PCOS Treatment**				
**Drugs**	**Mechanisms of Action**	**Indications**		**References**
Clomiphene	Selective inhibitor of estrogen receptors	Reduces estrogen levels and increases GnRH levels		[163]
Letrozole	Aromatase inhibitors	Decreases estrogen production		[164]
Gonadotrophins	Gonadotrophin hormone receptors stimulant	Increase gonadotrophin hormones release and induce ovulation		[165]
Acarbose	Reversibly inhibits α-glucosidase in the intestine and reduces the disaccharide and monosaccharide absorption	Decreased testosterone and very low-density lipoproteins		[166]
Metformin	Inhibition of hepatic glucose production via stimulation of AMP-activated protein kinase	Management of gestational diabetes, insulin resistance, obesity, and hyperglycemia. Metformin, in combination with statins, reduced C-reactive protein, total cholesterol, and low-density lipoprotein levels Combination of ursodeoxycholic acid and omega-3 fatty acids, and found ineffective for histological recovery of NASH.		[167,168,169]
Thiazolidinediones	Agonist for peroxisome proliferator-activated receptors	Relieve from NASH-associated PCOS symptoms such as IR and TAG levels		[170,171]
Inositol	Acts as a secondary messenger for insulin and improves glucose uptake	Improves menstrual irregularities, acne score, and IR		[172]
Oral Contraceptives	Various mechanisms such as decreasing follicle-stimulating hormones and luteinizing hormones levels	Manage menstrual and hirsutism		[173]
Eflornithine	Inhibits ornithine decarboxylase	Hirsutism		[174,175]
Piaglitazone and Vitamin E	PPARγ agonist and antioxidant	Significant recovery observed in NASH histological analysis		[130,176]
**Candidates in the Pipeline for NASH as Potential Alternative Treatment for PCOS**				
**Candidates**	**Mechanisms**	**Phase**	**Indications**	**References**
Elagolix	Low-molecular-weight non-peptide GnRH receptor antagonist	II	Reduces blood sex hormone levels and moderate to severe endometrium-associated pain.	[177]
Resmetriom	Thyroid hormone receptor β agonist	IIb and III	Effectively reduced hepatic fat in NASH patients; however, obesity may be the potential indicator in PCOS progression, so it can be effective in its treatment.	[178,179]
PXL770	AMPK agonist	Ib	Improves IR, NASH, and adrenoleukodystrophy	[180]
Tirzepatide	GLP-1 receptor activator	II	Reduces lipid content in NASH patients. GLP-1 stimulation is required for oocyte maturation so it might help manage PCOS ovulation issues.	[181,182,183]
Cotadutide	GLP-1 receptor stimulant	IIb and I (D.M)	Reduction in weight and blood glucose levels. Effective in NASH, DM, and may be for PCOS	[183,184,185]
Aramchol	Inhibits stearoyl-CoA desaturase 1	IIb	Showed markedly decreased lipid, ALT, and AST levels. It also improved HbA1c. Gestational diabetes and elevated sugar levels were observed in healthy women and PCOS patients. The abovementioned benefits may also provide relief to PCOS patients.	[186,187]
Belapectin	Galectin inhibitor	IIb	Reduced liver fat and was effective in treating liver fibrosis and portal hypertension. PCOS patients also experience hypertension and cholesterol issues, so it might be effective in PCOS.	[188,189]
Efruxifermin	Fibroblast growth factor 21 analog	IIa	Significant reduction in liver fibrosis by decreasing liver fat percentage, which can be used for managing lipid issues in PCOS patients.	[190]
Pegbelfermin	FGF21 analog	IIa	Reduced hepatic lipid content while patients experienced diarrhea	[191]
Lanifibranor	PPAR agonist	IIb	Showed a reduction in inflammatory markers of NASH patients; however, they experienced anemia and peripheral edema effects. It may be effective in PCOS because inflammatory mechanisms are involved in its progression.	[192]
VK2809	Thyroid hormone receptor β agonist	IIb	Alleviates hepatic fat content and prevents fibrosis. Considered as an alternative therapy for PCOS patients in managing lipids levels	[193,194,195]
CER-209	Agonist of P2Y13 GPCR and acts via reverse lipid transport pathway	I	Recovery from NASH by showing a marked reduction in TAG and cholesterol levels. P2Y receptor levels were significantly reduced, so it may be effective in PCOS treatment.	[196,197]

## Data Availability

This manuscript has no associated data.

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
