# Peer review of "Dysregulated Liver Metabolism and Polycystic Ovarian Syndrome"

_ijms, 2023, doi:10.3390/ijms24087454_

Round 1
Reviewer 1 Report
Reviewer’s Comments:
The manuscript “Emerging Metabolic Mechanisms on Polycystic Ovarian Syndrome” is a very interesting work. In this work, a significant fraction of couples around the world suffers from polycystic ovarian syndrome (PCOS), a disease defined by the characteristics of enhanced androgen synthesis in ovarian theca cells, hyperandrogenemia, and ovarian dysfunction in women. Most of the clinically observable symptoms and altered blood biomarker levels in the patients indicate metabolic dysregulation and adaptive changes as the key underlying mechanisms. Since the liver is the metabolic hub of the body and is involved in steroid-hormonal detoxification, pathological changes in the liver may contribute to female endocrine disruption, potentially through the liver-to-ovary axis. Of particular interest are hyperglycemic challenges and the consequent changes in liver-secretory protein (s) and insulin sensitivity affecting the maturation of ovarian follicles, potentially leading to female infertility. While I believe this topic is of great interest to our readers, I think it needs major revision before it is ready for publication. So, I recommend this manuscript for publication with major revisions.
1. In this manuscript, the authors did not explain the importance of the Metabolic Mechanisms the introduction part. The authors should explain the importance of Metabolic Mechanisms.
2) Title: The title of the manuscript is not impressive. It should be modified or rewritten it.
3) Correct the following statement “The purpose of this review is to provide insight into emerging metabolic mechanisms underlying PCOS as the primary culprit, which promotes its incidence and aggravation of PCOS, and to summarize medications and new potential therapeutic approaches for the disease”.
4) Keywords: The Metabolic Mechanisms is missing in the keywords. So, modify the keywords.
5) Introduction part is not impressive. The references cited are very old. So, Improve it with some latest literature such as 10.3390/molecules27217368, 10.3390/molecules27207129
6) The authors should explain the following statement with recent references, “TLR4 on the Kupffer cell (KC) membrane stimulates the production of the cytokines responsible for inflammation”.
7) Add space between magnitude and unit. For example, in synthesis “21.96g” should be 21.96 g. Make the corrections throughout the manuscript regarding values and units.
8) The author should provide reason about this statement “Because of the compromised liver, there might be no proper cure for PCOS patients except for its management”.
9. Comparison of the present results with other similar findings in the literature should be discussed in more detail. This is necessary in order to place this work together with other work in the field and to give more credibility to the present results.
10) Conclusion part is very long. Make it brief and improve by adding the results of your studies.
11) There are many grammatic mistakes. Improve the English grammar of the manuscript.
Author Response
Reviewer #1
The manuscript "Emerging Metabolic Mechanisms on Polycystic Ovarian Syndrome" is a very interesting work. In this work, a significant fraction of couples around the world suffers from polycystic ovarian syndrome (PCOS), a disease defined by the characteristics of enhanced androgen synthesis in ovarian theca cells, hyperandrogenemia, and ovarian dysfunction in women. Most of the clinically observable symptoms and altered blood biomarker levels in the patients indicate metabolic dysregulation and adaptive changes as the key underlying mechanisms. Since the liver is the metabolic hub of the body and is involved in steroid-hormonal detoxification, pathological changes in the liver may contribute to female endocrine disruption, potentially through the liver-to-ovary axis. Of particular interest are hyperglycemic challenges and the consequent changes in liver-secretory protein (s) and insulin sensitivity affecting the maturation of ovarian follicles, potentially leading to female infertility. While I believe this topic is of great interest to our readers, I think it needs major revision before it is ready for publication. So, I recommend this manuscript for publication with major revisions.
- In this manuscript, the authors did not explain the importance of the Metabolic Mechanisms in the introduction part. The authors should explain the importance of Metabolic Mechanisms.
Answer: The authors thank the reviewer for the valuable comment. We incorporated the portion of the metabolic mechanism in the introduction. The liver is the metabolic hub of the body because all ingested nutrients pass through the liver after intestinal absorption. Therefore, underlying molecular cascades of metabolic diseases include lipotoxicity, autophagy dysregulation, endoplasmic reticulum stress, IR, and other targets. NAFLD is currently the most common liver disease globally and has been reported to affect 30% of people over the age of 18. Another study showed that NAFLD is strongly associated with obesity and T2D [1].
- Title: The title of the manuscript is not impressive. It should be modified or rewritten it.
Answer: The authors appreciate the helpful comment. We have updated the title of the article.
- 3. Correct the following statement "The purpose of this review is to provide insight into emerging metabolic mechanisms underlying PCOS as the primary culprit, which promotes its incidence and aggravation of PCOS, and to summarize medications and new potential therapeutic approaches for the disease". Keywords: The Metabolic Mechanisms is missing in the keywords. So, modify the keywords.
Answer: The authors would like to thank the reviewer. We have added the word metabolic mechanisms in the keywords.
- Introduction part is not impressive. The references cited are very old. So, Improve it with some latest literature such as 10.3390/molecules27217368, 10.3390/molecules27207129
Answer: The authors thank the reviewer for their valuable comments regarding the references. We incorporated the latest citations in the introduction part. Unfortunately, the references suggested by you do not fit into the scope of our review because our review is majorly focused on PCOS and its association with liver metabolic abnormalities and mechanisms oriented based on animal studies and patient-based studies. Drugs are included in relevance with clinical trials, while suggested articles are not related to PCOS and liver metabolic disorders. However, we cited the second reference based on a brief discussion regarding type 2 diabetes and insulin resistance. However, PCOS has been reclassified as a metabolic disorder since insulin resistance (IR) is acknowledged as a key pathogenic parameter [2, 3]. Patients with PCOS commonly have a high prevalence of obesity and IR [2], carrying an increased risk of metabolic syndrome, T2D, and cardiovascular disease [4, 5].
- The authors should explain the following statement with recent references, "TLR4 on the Kupffer cell (KC) membrane stimulates the production of the cytokines responsible for inflammation".
Answer: The authors appreciate the reviewer's meticulous reading of our manuscript. TLR4 on the Kupffer cell (KC) membrane stimulates the production and release of the cytokines responsible for inflammation. TLR4 stimulation triggers p38, MAPK, PI3K, and NF-kB, followed by enhanced cytokine production [6]. This was included in the revised MS.
- Add space between magnitude and unit. For example, in synthesis "21.96g" should be 21.96 g. Make the corrections throughout the manuscript regarding values and units.
Answer: The authors would like to thank the reviewer for the suggestion. We added the space between magnitude and unit.
- The author should provide reason about this statement "Because of the compromised liver, there might be no proper cure for PCOS patients except for its management".
Answer: Studies have proven that due to compromised liver results in NAFLD, there are alterations in SHBG, hyperinsulinemia, ALT and AST activities, and testosterone, which must be managed to cure PCOS. In the revised MS, we suggest that drugs currently in the pipeline can be used as alternative treatment options for managing PCOS symptoms
- 8. Comparison of the present results with other similar findings in the literature should be discussed in more detail. This is necessary in order to place this work together with other work in the field and to give more credibility to the present results.
Answer: In our manuscript, we already have shown the comparison of various studies in support of our manuscript. In the case of NAFLD, the compromised liver leads to decreased androgen metabolism and increased metabolism of SHBG, which propagate through different pathways and cause HA, hyperinsulinemia, and resultant infertility [7]. Moreover, many studies have reported increased ALT and AST activities in PCOS patients [8]. Previous experimental studies and clinical profiling of patients have proven that variations in BMI are directly associated with plasma SHBG levels in men and women [9]. The reduced SHBG and IGF, preferentially insulin-like growth factor-binding protein 1, have been reported to be associated with hyperinsulinemia and represented as critical biomarkers/indicators of metabolic syndrome [10]. The results of the meta-analysis comprising compromised hepatic diseases reveal that SHBG levels significantly decreased in women, whereas serum total testosterone levels increased [11]. Studies have proven that reduced SHBG levels, HA, hyperinsulinemia, and T2D are the primary symptoms of PCOS [12]. Thus, increased levels of testosterone are indicative of PCOS and may also be associated with NAFLD. The results of the meta-analysis and systemic review of 17 studies comprising 2734 PCOS patients and 2561 healthy and BMI-matched women reveal that PCOS women who also have NAFLD exhibited increased serum testosterone levels and an increased free androgen index compared with those who did not have NAFLD [13].
A retrospective longitudinal cohort study was done in the United Kingdom using the primary care database, which included 63,120 NAFLD women suffering from PCOS and 121,064 healthy and BMI-matched women from January 2000 to May 2016. The results showed that women with serum testosterone >3.0 nmol/L were 2.3 times more likely to be diagnosed with NAFLD than those who had normal testosterone levels [hazard ratio (HR) =2.30, 95% confidence interval (CI) 1.16-4.53, p=0.017]. Also, women with SHBG < 30 nmol/L were 4.75 times more likely to be diagnosed with NAFLD than those with normal SHGB levels (HR = 4.75, 95% CI 2.44-9.25, p<0.001) [14]. A case-control study including 29 young PCOS patients and 22 healthy controls used proton-magnetic resonance spectroscopy to confirm that women with PCOS have increased levels of hepatic fat compared to control individuals, confirming that PCOS patients have a risk of NAFLD [15]. Another case-control study, including 275 serum samples of PCOS patients selected under the Rotterdam criteria and analyzed by liquid chromatography-mass spectrometry (LC-MS), showed that PCOS patients had increased testosterone, dihydrotestosterone, fasting glucose, ALT, and AST levels, which are clearly indicative of hepatic disease [16].
- Conclusion part is very long. Make it brief and improve by adding the results of your studies.
Answer: We have shortened the conclusion portion as per your suggestion.
- There are many grammatic mistakes. Improve the English grammar of the manuscript.
Answer: The English company's editor checked the manuscript's grammar.
I appreciate your kind final consideration for disseminating our review paper in the International Journal of Molecular Sciences.
Sincerely,
Sang Geon Kim, Ph.D., Professor, and Dean
College of Pharmacy
Dongguk University-Seoul
Republic of Korea;
Tel: +8231-961-5218; Fax: +8231-961-5206; E-mail: sgkim@dongguk.edu; sgk@snu.ac.kr
References of the answers
- Khan, M. S.; Lee, C.; Kim, S. G. Non-alcoholic fatty liver disease and liver secretome. Arch Pharm Res 2022, 1-26. https://doi.org/10.1007/s12272-022-01419-w
- Khan, S.; Iqbal, S.; Shah, M.; Rehman, W.; Hussain, R.; Rasheed, L.; Alrbyawi, H.; Dera, A. A.; Alahmdi, M. I.; et al. Synthesis, In Vitro Anti-Microbial Analysis and Molecular Docking Study of Aliphatic Hydrazide-Based Benzene Sulphonamide Derivatives as Potent Inhibitors of α-Glucosidase and Urease. Molecules 2022, 27, (20), 7129. https://doi.org/10.3390/molecules27207129
- Kim, J. J.; Hwang, K. R.; Oh, S. H.; Chae, S. J.; Yoon, S. H.; Choi, Y. M. Prevalence of insulin resistance in Korean women with polycystic ovary syndrome according to various homeostasis model assessment for insulin resistance cutoff values. Fertil. Steril. 2019, 112, (5), 959-966 e1. https://doi.org/10.1016/j.fertnstert.2019.06.035
- Hoeger, K. M.; Dokras, A.; Piltonen, T. Update on PCOS: consequences, challenges, and guiding treatment. J Clin Endocrinol Metab 2021, 106, (3), e1071-e1083. https://doi.org/10.1210/clinem/dgaa839
- Franik, G.; Bizoń, A.; Szynkaruk-Matusiak, M.; Osowska, K.; Dryś, A.; Olszanecka-Glinianowicz, M.; Madej, P. The association between 24-hour ambulatory blood pressure measurement and selected biochemical and anthropometric parameters in women with polycystic ovary syndrome. Eur Rev Med Pharmacol Sci 2021, 25, (11), 3947-3954. https://doi.org/10.26355/eurrev_202106_26035
- Schwabe, R. F.; Seki, E.; Brenner, D. A. Toll-like receptor signaling in the liver. Gastroenterology 2006, 130, (6), 1886-1900. https://doi.org/10.1053/j.gastro.2006.01.038
- Di Stasi, V.; Maseroli, E.; Rastrelli, G.; Scavello, I.; Cipriani, S.; Todisco, T.; Marchiani, S.; Sorbi, F.; Fambrini, M.; et al. SHBG as a Marker of NAFLD and Metabolic Impairments in Women Referred for Oligomenorrhea and/or Hirsutism and in Women With Sexual Dysfunction. Front. Endocrinol. (Lausanne) 2021, 12. https://doi.org/10.3389/fendo.2021.641446
- Wilson, C. High risk of liver disease in women with polycystic ovary syndrome. Nat. Rev. Endocrinol. 2010, 6, (3), 122-122. https://doi.org/10.1038/nrendo.2009.284
- Davidson, B.; Gambone, J.; Lagasse, L.; Castaldo, T.; Hammond, G.; Siiteri, P.; Judd, H. Free estradiol in postmenopausal women with and without endometrial cancer. J. Clin. Endocrinol. Metab. 1981, 52, (3), 404-407. https://doi.org/10.1210/jcem-52-3-404
- Tchernof, A.; Toth, M. J.; Poehlman, E. T. Sex hormone-binding globulin levels in middle-aged premenopausal women. Associations with visceral obesity and metabolic profile. Diabetes Care 1999, 22, (11), 1875-1881. https://doi.org/10.2337/diacare.22.11.1875
- Wildman, R. P.; Wang, D.; Fernandez, I.; Mancuso, P.; Santoro, N.; Scherer, P. E.; Sowers, M. R. Associations of testosterone and sex hormone binding globulin with adipose tissue hormones in midlife women. Obesity 2013, 21, (3), 629-636. https://doi.org/10.1002/oby.20256
- Selva, D. M.; Hogeveen, K. N.; Innis, S. M.; Hammond, G. L. Monosaccharide-induced lipogenesis regulates the human hepatic sex hormone–binding globulin gene. J. Clin. Invest. 2007, 117, (12), 3979-3987. https://doi.org/10.1172/JCI32249
- Rocha, A.; Faria, L.; Guimarães, T.; Moreira, G.; Cândido, A.; Couto, C.; Reis, F. Non-alcoholic fatty liver disease in women with polycystic ovary syndrome: systematic review and meta-analysis. J. Endocrinol. Invest. 2017, 40, (12), 1279-1288. https://doi.org/10.1007/s40618-017-0708-9
- Kumarendran, B.; O'Reilly, M. W.; Manolopoulos, K. N.; Toulis, K. A.; Gokhale, K. M.; Sitch, A. J.; Wijeyaratne, C. N.; Coomarasamy, A.; Arlt, W.; et al. Polycystic ovary syndrome, androgen excess, and the risk of nonalcoholic fatty liver disease in women: A longitudinal study based on a United Kingdom primary care database. PLoS Med. 2018, 15, (3), e1002542. https://doi.org/10.1371/journal.pmed.1002542
- Jones, H.; Sprung, V. S.; Pugh, C. J.; Daousi, C.; Irwin, A.; Aziz, N.; Adams, V. L.; Thomas, E. L.; Bell, J. D.; et al. Polycystic ovary syndrome with hyperandrogenism is characterized by an increased risk of hepatic steatosis compared to nonhyperandrogenic PCOS phenotypes and healthy controls, independent of obesity and insulin resistance. J. Clin. Endocrinol. Metab. 2012, 97, (10), 3709-3716. https://doi.org/10.1210/jc.2012-1382
- Münzker, J.; Hofer, D.; Trummer, C.; Ulbing, M.; Harger, A.; Pieber, T.; Owen, L.; Keevil, B.; Brabant, G.; et al. Testosterone to dihydrotestosterone ratio as a new biomarker for an adverse metabolic phenotype in the polycystic ovary syndrome. J. Clin. Endocrinol. Metab. 2015, 100, (2), 653-660. https://doi.org/10.1210/jc.2014-2523

Reviewer 2 Report
In this review, Khan et al. describe the pathology and medications in use or under development for PCOS treatment. The strengths of the manuscript are the general introduction of the pathology of PCOS, including sex hormone dysregulation, IR, oxidative stress, etc., and the summary of diverse medications, including the mechanisms of drug action and indications. The weaknesses lie in: 1. the lack of novelty, especially the pathology section and the two figures; 2. the accuracy of some expression; 3. new drugs under development mainly focused on NAFLD, which is only one complication of PCOS. In addition, the title of the review is misleading as the content related to metabolic mechanisms is less than other details. There is also one obvious mistake in abbreviations at the end of the review.
Author Response
Reviewer #2
In this review, Khan et al. describe the pathology and medications in use or under development for PCOS treatment. The strengths of the manuscript are the general introduction of the pathology of PCOS, including sex hormone dysregulation, IR, oxidative stress, etc., and the summary of diverse medications, including the mechanisms of drug action and indications. The weaknesses lie in: 1. the lack of novelty, especially the pathology section and the two figures; 2. the accuracy of some expression; 3. new drugs under development mainly focused on NAFLD, which is only one complication of PCOS. In addition, the title of the review is misleading as the content related to metabolic mechanisms is less than other details. There is also one obvious mistake in abbreviations at the end of the review.
- The lack of novelty, especially the pathology section and the two figures
Answer: The author would like to thank the reviewer for their generous insight. The pathology section of our manuscript tries to link two different aspects, i.e., liver metabolic disorder and PCOS, together. Currently, study data is less available, particularly on patient-based studies, because of the inter-related complications. However, figure 1 of the manuscript is based on the data we included in the manuscript, while Figure 2 panel B is different from the other figures available on various databases. In panel B of Figure 2, we discussed the potential effect of ITIH1 on ovarian cells by linking it to the liver. Reduced levels of Gα13 in the liver stimulate hypersecretion of ITIH1 from the liver. Circulating ITIH1 may travel to the ovarian theca cells and granulosa cells. The binding of ITIH1 to hyaluronic acid on the cell surface may prevent insulin from binding with its receptors.
- The accuracy of some expression
Answer: The authors would like to appreciate the reviewers' efforts to study our manuscript in detail. We have corrected the expression throughout the manuscript.
- New drugs under development mainly focus on NAFLD, which is only one complication of PCOS. In addition, the title of the review is misleading as the content related to metabolic mechanisms is less than other details. There is also one obvious mistake in abbreviations at the end of the review.
Answer: The authors thank the reviewer for the helpful comments. The dilemma of the manuscript is based on establishing the relationship between liver metabolic disorder and PCOS. The drugs currently available to discover potential PCOS treatment are focused (i.e., treatment of complications like obesity and hyperglycemia). PCOS is related to an endocrine disorder, and because of that, it is difficult to establish in animals and proceed with clinical studies. Studies have shown that liver diseases such as NAFLD are associated with obesity and hyperglycemia. For the above-mentioned reasons, we suggested the NAFLD drugs in the pipeline as an alternative treatment for managing the symptoms. We also edited the title of the manuscript.
I appreciate your kind final consideration for disseminating our review paper in the International Journal of Molecular Sciences.
Sincerely,
Sang Geon Kim, Ph.D., Professor, and Dean
College of Pharmacy
Dongguk University-Seoul
Republic of Korea;
Tel: +8231-961-5218; Fax: +8231-961-5206; E-mail: sgkim@dongguk.edu; sgk@snu.ac.kr

Reviewer 3 Report
Dear Authors,
Thank you for allowing me to review your manuscript.
The study is very interesting as it offers a comprehensive overview of the etiology of PCOS.
I would like it to be focused in the study of possible pharmacological and clinical developments on the basis of this new knowledge.
Finally, for a multidisciplinary approach we propose to mention:
DOI: 10.1186/s13027-022-00465-9
DOI: 10.3390/jpm12091536
Doi: 10.1016/S0140-6736(20)30068-4
Doi: 10.1002/jmv.28208
Author Response
Reviewer 3 Comments
Thank you for allowing me to review your manuscript. The study is very interesting as it offers a comprehensive overview of the etiology of PCOS. I would like it to be focused in the study of possible pharmacological and clinical developments on the basis of this new knowledge. Finally, for a multidisciplinary approach we propose to mention:
DOI: 10.1186/s13027-022-00465-9
DOI: 10.3390/jpm12091536
Doi: 10.1016/S0140-6736(20)30068-4
Doi: 10.1002/jmv.28208
Answer: The authors would like to thank reviewer for the helpful suggestions. We included the first two references in our manuscript. Microbial [1] or endogenous cytokine production results in the stimulation of NLRP3 and pro-interleukin-1β via NF-kB. The findings of another study suggest that supplementation of Myo-Inositol (1200 mg per day) and D-Chiro-Inositol (135 mg per day) for 12 months significantly improved FSH, LH, and progesterone as well as oligomenorrhea, and follicle count [2]. Unfortunately, the other two references do not fit in the scope of our manuscript.
I appreciate your kind final consideration for disseminating our review paper in the International Journal of Molecular Sciences.
Sincerely,
Sang Geon Kim, Ph.D., Professor, and Dean
College of Pharmacy
Dongguk University-Seoul
Republic of Korea;
Tel: +8231-961-5218; Fax: +8231-961-5206; E-mail: sgkim@dongguk.edu; sgk@snu.ac.kr
References of the answers
- Dellino, M.; Cascardi, E.; Laganà, A. S.; Di Vagno, G.; Malvasi, A.; Zaccaro, R.; Maggipinto, K.; Cazzato, G.; Scacco, S.; et al. Lactobacillus crispatus M247 oral administration: Is it really an effective strategy in the management of papillomavirus-infected women? Infect Agents Cancer 2022, 17, (1), 1-8. https://doi.org/10.1186/s13027-022-00465-9
- Dellino, M.; Cascardi, E.; Leoni, C.; Fortunato, F.; Fusco, A.; Tinelli, R.; Cazzato, G.; Scacco, S.; Gnoni, A.; et al. Effects of Oral Supplementation with Myo-Inositol and D-Chiro-Inositol on Ovarian Functions in Female Long-Term Survivors of Lymphoma: Results from a Prospective Case–Control Analysis. J Pers Med 2022, 12, (9), 1536.

Round 2
Reviewer 2 Report
I have no further comments.